# Reconsidering Leadership Development: From Programs to Developmental Systems

**DOI:** 10.3390/bs14070548

**Published:** 2024-06-28

**Authors:** David V. Day, Laura Dannhäuser

**Affiliations:** Kravis Leadership Institute, Claremont McKenna College, Claremont, CA 91711, USA; laura.dannhaeuser@claremontmckenna.edu

**Keywords:** leadership development, leader development, leadership development programs, open systems theory, systems thinking

## Abstract

We argue for reconsidering leadership development based on open systems theory and systems design principles. A primary advantage of open systems thinking is that it encourages holistic approaches to development and avoids episodic program-based training and piecemeal thinking. The latter approaches are both limited and limiting yet tend to be the preferred approach to leadership development in organizations. Open systems approaches to development offer numerous advantages both conceptually and pragmatically, especially through the incorporation of ongoing feedback cycles. Core practices that define a leadership development system are presented and implications are discussed.

## 1. Introduction

In rethinking leadership development, which is the focus of this Special Issue, a question to be addressed is why it needs to be rethought. Rethinking implies that present approaches to leadership development have reached their limits in terms of developmental impact. We agree with that assessment; however, perhaps a different way of addressing this issue is to reconsider leadership development based on what we know from a scientific, evidence-based perspective. If we approached leadership development from the perspective of science, how would that motivate a reconsideration? Rethinking implies a new start with a clean state that ignores or rejects what has been established. A reconsideration tries to avoid the problem of “throwing out the baby with the bathwater” by retaining the good (i.e., effective) aspects of extant leadership development practices while considering novel approaches to certain intractable problems.

A case was made more than 20 years ago that the field of leadership development was almost completely co-opted by practice perspectives [1]. The point was also made that whereas leadership development is big business, the evidence for its effectiveness was weak. Two things have not changed. The first is that leadership development is still a big business. Investments globally run into the billions of dollars and there is a plethora of consultancies that offer leadership development “solutions”. The claim of solutions suggests strongly that there are proven answers to the various challenges of developing leaders and leadership. The second aspect that has not changed over the previous two decades is that the evidence for effectiveness of these so-called solutions is hardly compelling. Another way of stating this is that the evidence for effective solutions to various problems or challenges of leadership development is weak.

No less an expert on organizational leadership than McKinsey & Company has noted both the importance and the shortcomings of contemporary initiatives on leadership development. McKinsey researchers conducted interviews with more than 500 senior executives and found that approximately two thirds identified leadership development as their most urgent concern [2]. This attests to the importance of the underlying issues associated with the development of leadership talent. McKinsey researchers also reported in that article that fewer than 10 percent of senior managers think their companies develop global leaders effectively, and about one third of US companies lament they do not have enough leaders with the right capabilities. Although this might not be the most scientific evidence to inform the issue, these data come from those with a close-hand perspective on leadership needs in corporate for-profit organizations. The perspectives captured in the McKinsey research suggest that a great deal of the investments in leadership development is probably wasted or at least not invested wisely in the most effective strategies to address their leadership (development) needs. We are aware of no more recent data that suggest this situation has changed for the better.

There is a saying attributed to Albert Einstein that insanity is doing the same thing over and over while expecting different results (https://www.scientificamerican.com/article/einstein-s-parable-of-quantum-insanity/ accessed on 23 July 2023). Applying this to the leadership development industry suggests that the whole field is a bit crazy, which, when you think about the amount of money involved relative to documented impact, has a ring of truth to it. But what is the alternative? One approach might be to think about leadership development in a more holistic systems manner than by the “insane” (i.e., ineffective) approach associated with leadership programs as the solution to the leadership development conundrum in organizations.

Something else to consider is that programs are a commodity and undoubtedly the largest and most expensive commodity in the marketplace of leadership development. There are advantages to packaging development interventions as programs because it makes them conducive for marketing and sales as discrete products. Leadership development programs typically include costs associated with assessments (usually proprietary), facilitation, and overhead such as classroom space, meals, and lodging (if the program is residential). The rest is pure gravy for the vendor. These costs are bundled into an overall program expense, which is relatively easy to justify against something like an HR budget given that is there is typically an annual amount to invest in development for most corporate organizations.

Despite certain advantages such as the widespread acceptance of programs as the appropriate way to conduct leadership development, there are distinct disadvantages. A major limitation with programs is their bounded and episodic nature. Even the best programs have a relatively limited time scale that does not match with the underlying nature of development in general and leadership development more specifically [3,4]. Developing leaders and leadership can take years or even decades, and typically involves a large commitment to ongoing deliberate practice. A program is usually a few hours, a few days, or perhaps a few weeks. There might be exceptions to this general rule, but those would be the few exceptions that prove the rule.

A different way to consider or think about the design and delivery of leadership development initiatives (i.e., interventions) is as systems. One advantage of systems thinking when it comes to leadership development is that this type of thinking identifies the various components but extends a step beyond that in specifying how the components are interrelated to the whole. Systems thinking is based on a more holistic perspective, in this case, about leadership development. Although there has been some attention in the literature to the notion of leader and leadership development systems [5,6], we take a different perspective on systems by grounding our ideas in open systems theory.

## 2. Open Systems Theory

One of the most renown theories in the organizational sciences is that of open systems theory [7]. The authors of open systems theory proposed a model for understanding any organization as “an energic input–output system in which the energic return from the output reactivates the system” (p. 20). The theory is grounded in the characteristic state of the living organism, which can be compared to a closed system in this way:

“A system is closed if no material enters or leaves it; it is open if there is import and export and, therefore, change of components. Living systems are open systems, maintaining themselves in exchange of materials with environment, and in continuous building up and breaking down of their components”[8] (p. 23)

As is discussed, this makes for a compelling way to view leadership development initiatives as a transformation of energy into outputs to the environment that elicit feedback to be used as input to reactivate the system. In the case of leadership, that system is a developing individual or broader collective. A key word in the description by von Bertalanffy—considered the founder of open systems theory—is “continuous” in terms of building up and breaking down of the components.

In essence, there is no openness to the environment with a closed system. As a result, there are no external resources returned to reactivate the system. There is also no continuous building up and breaking down of systems attributes (i.e., no development). Reconsidering leadership development involves reconsidering how to reinvent or reconceptualize it in terms of open systems principles.

According to Katz and Kahn, the following ten characteristics are descriptive of all open systems (see Table 1 for a summary). With each of these principles, a brief explanation is provided as to how each is potentially relevant to a leadership development system. A primary point is if organizations can be open (i.e., “living”) systems, then why not a leadership development initiative? What would it take to reconsider leadership development in systems terms rather than as programs? The core characteristics of open systems are:Importation of energy: The energy in a leadership development system comes from the motivation to develop. These characteristics might also apply to leadership development programs, but sometimes individuals are sent to programs even though there is no demonstrated motivation to develop as a leader.Throughput or transformation: Energy is transformed into some sort of new learning with the acquisition of new skills or mindsets changing in ways that involve some reorganization of the input. This might also be characteristic of a leadership development program. Indeed, it might be the most defining characteristic of a program.Output: There is something exported into the environment. This could be in the form of new ways of thinking, new behaviors, or deliberate practice to further develop leadership skills and competencies. There might be some of this as follow-up to a program, but there are few if any guarantees that program participants engage in such post-program activities. Participants are “released” as output into the respective organizational environments, but there is no assurance that what was developed in a program is also released. There is the saying that sending a changed person back into an unchanged system often is an exercise in futility. There is little in the way of support from a system guided by unchanged principles.Systems as cycles of events: The pattern of activities that define the energy exchange have a cyclical nature. If leadership development is conceptualized as ongoing development practices in everyday life, then the cyclical exchange is clear. There is no return of energy following the completion of a program. Once the program ends, so does the system in which it was created.Negative entropy: The entropic process is “a universal law of nature in which all forms of organizations move toward disorganization or death” [7] (p. 25). Open systems can reverse or delay the natural entropic progression through the exchange of energy from the environment. Closed systems do not have this capability, and that includes leadership development programs. There is a set date at which a program ends (i.e., dies). Continuing lifelong learning and development as a leader is a way of staying or delaying that terminal state.Information input, negative feedback, and the coding process: Information serves as an environmental input in addition to energy. The feedback a developing leader receives in the environment is one such information input into the cycle. In comparison, feedback in a leadership program often takes the form of 360-degree or multisource feedback reports, which are akin to driving a car while looking in the rearview mirror. These reports are feedback about past behavior or events and not an ongoing or continuous process grounded in environmental inputs.Steady state and dynamic homeostasis: When thinking about a steady state, it is important to consider that this state is steady in a dynamic way, like a gyroscope. But given that the nature of leadership development is change, it needs to be considered not as unchanging but as preservation of the character of the system as a developmental system. Although there is development and therefore change is occurring, the fundamental nature of the leadership development system is a commitment to ongoing development. This is why self-views are so important in long-term development, especially in terms of internalizing a leader identity [9]. People devote their time to what they value most. What is valued is a reflection of one’s identity.Differentiation: Different leadership skills and self-views that are developed as part of leader development can be considered as different strands in an overall web of individual development [10]. These differentiated strands or skills and self-views develop in their own way independent of each other, at least initially. An important consideration in systems design is that differentiation always precedes integration [11].Integration and coordination: Consider the web of development again. As differentiation proceeds, it is supported by processes that bring the system together for more holistic or unified functioning. But a key issue with integration and coordination is time. It takes time to build on differentiation of leadership skills and self-views to integrate and coordinate at a more holistic level. Most leadership development programs are bounded in terms of relatively short-term time frames.Equifinality: People start at different places and change in different ways on their developmental journeys. This is especially the case when an individual owns their development by choosing what to develop and how to do it. A system in the form of a developing leader can reach the same final state from differing initial conditions and by a variety of pathways. Most programs adopt either implicitly or explicitly a training approach in which all participants follow a set of structured practices in a set order with the goal of having all participants end the program at the same level of standard [12].

A major difference between an open and a closed system is that an open system has permeable boundaries that allows interaction with the environment. An open system is characterized by cyclical events that transform inputs in terms of energy and information (as well as developing skills and self-views) to outputs into the environment in the form of behavioral practice with feedback about that practice returned as input to the system. An essential characteristic of an open system is that it comprises repeating cycles of events and is not a single, stand-alone episode like a program (see Figure 1).

The cycle of repeating events in developmental systems operates in two ways. One cycle involves the transfer of developmental energy in the form of motivation into leadership skills and self-views. These transformed resources provide the outputs released into the environment in the form of leadership behaviors and sensemaking filters. These outputs are the foundation for practice that drive ongoing leader development by experiential learning through doing. From this practice comes feedback, either formal or informal. Did things go well or not so well? What was learned from engaging in or practicing a leadership skill or mindset? Information is imported from the practice environment or experience, which forms the basis of inputs back to the system for the next cycle of development. This feedback mechanism is one of the core characteristics of an open system in which new information also is fed back to the system that contributes to building energy or motivation for further developmental transformation.

Another defining characteristic of open systems is equifinality, which maintains that any system can reach the same final state from differing initial conditions and through any number of different paths. In terms of developmental systems, this characteristic highlights the true nature of development in terms of how individuals start their respective developmental journeys at different places and change in different ways. This is a key aspect of personal trajectories of development that has been adopted in the study of leader development [13]. It is also the case that the focus of the development differs across individuals because of individual differences in leadership strengths and weaknesses. Two different individuals can choose two very different skills or mindsets to work on to become more effective leaders. With training, it is assumed that individuals all start pretty much at the same place and change in the same way over the course of the training. These assumptions do not hold when the focus is development because it is assumed that individuals start at different developmental levels and have different developmental trajectories for a given skill or self-view.

## 3. From Program-Based to Systems Approaches in Leadership Development

Rethinking leadership development through the lens of systems design principles grounded in an evidence-based perspective involves a more holistic and integrative approach that contrasts with traditional leadership training and development methods. This approach is built on the premise that leadership is not just about individual capabilities but also about the systems within which leaders operate. Transforming leadership development involves moving from periodically attending a highly structured program to an ongoing long-term perspective on self-authored and personalized development.

The point was made at the opening of this article that reconsidering leadership development does not mean completely rethinking it. State-of-the-art leadership development involves a few systems-based principles, especially those practices that take participants out of the classroom and leverage ongoing work for development. But it is still the case that too much emphasis is placed on program-based interventions. A major challenge to such approaches is that they tend to operate (implicitly) as closed systems, especially in not having continuous and ongoing inputs from the environment.

The following points provide additional background material on systems thinking and design beyond what is considered in open systems theory and is focused more tightly on leadership development. There has been a great deal of interdisciplinary work on systems theory since the Katz and Kahn contributions to the social psychology of organizations (see [14] for more detail on why systems thinking is important). It is not possible to include all the different systems principles that define the field; however, the following represent some of the most relevant ones to consider in moving the field of leadership development to a more systems-based approach.

### 3.1. Understanding Systems Thinking

State-of-the-art leadership development involves a holistic view on development. Systems thinking encourages leaders to view organizations as complex, interrelated structures and processes, rather than collections of independent and mutually exclusive parts. This means understanding how different elements within an organization interact and influence each other [15,16]. The notion of understanding the various components in a system as well as the nature of the interdependencies between them is the basis for one definition of complexity [17]. Taking an evidence-based approach to understanding complex systems means that decisions are made based on data and evidence rather than intuition or tradition [18,19]. Being evidence-based involves collecting, analyzing, and using data to understand system dynamics, identify leverage points, and inform interventions. The following constitute additional considerations in understanding and implementing leadership development using the principles of complex systems and systems thinking.

### 3.2. Leveraging Feedback Loops

Identifying and understanding feedback loops within an organization can help leaders anticipate and manage the consequences of their actions. A positive feedback loop occurs when the output of a system amplifies the system’s performance or accelerates a process [20]. This does not necessarily mean positive in terms of valence or desirability; rather, it refers to the direction of change. A key characteristic of a positive feedback loop is that it increases or enhances the rate of a process or change within a system. Thus, positive feedback loops lead to exponential growth or decline, driving systems away from equilibrium and potentially causing runaway effects.

A negative feedback loop occurs when the output of a system results in a counteraction that diminishes the effects of a change, promoting stability or equilibrium [21]. The negative aspect refers to the reduction in or negation of the change, not to the desirability of the outcome. Thus, negative feedback loops are stabilizing forces in systems, helping to moderate changes and maintain balance. They work to bring a system back to its set point or equilibrium, resisting deviations caused by external or internal changes. In short, positive feedback loops amplify changes, while negative feedback loops attempt to maintain dynamic stability (i.e., steady state or dynamic homeostasis). Both positive and negative feedback loops are ongoing and dynamic processes in systems thinking.

### 3.3. Adopting a Multi-Level Perspective

State-of-the-art leadership development involves perspectives at the individual, team, organizational-level, as well as cross-level interactions [22]. As such, effective leadership development based on systems design principles address competencies at multiple levels. This includes developing personal leadership qualities while also enhancing team dynamics and shaping organizational culture [23]. Understanding how changes at one level affect other levels is crucial. Evidence-based practices help in designing interventions that are coherent across different levels of the system [24].

### 3.4. Emphasizing Adaptability and Resilience

Organizations can be conceptualized as complex adaptive systems that are constantly evolving [25]. Leadership development should therefore focus on enhancing adaptability and resilience. Evidence-based learning and adaptation become the norm. Leaders are encouraged to experiment, learn from outcomes, and adapt strategies accordingly [26]. This involves systematically gathering and analyzing evidence about the effectiveness of different approaches.

### 3.5. Fostering Innovation and Creativity

Incorporating design thinking into leadership development encourages innovation and a user-centric approach to problem-solving [27]. Creativity fuels ongoing development. Based on evidence, leaders learn to iterate, prototype, and test solutions in real-world settings, embracing failure as a learning opportunity.

### 3.6. Building Collaborative Networks

Recognizing that no leader or organization operates in isolation, developing systems-oriented leaders involves building skills for cross-functional and cross-organizational collaboration [28]. Sharing data, insights, and best practices across networks helps in leveraging collective intelligence and amplifying leader and organizational impact. Implementing these principles requires a shift in how leadership development is designed, delivered, and evaluated.

Curriculum Design. Integrating systems thinking and design thinking principles into the long-term development curriculum enables leaders to see the bigger picture, understanding the complex interdependencies within and outside their organizations. Design thinking adds a human-centered approach to problem-solving, encouraging creativity and innovation. Together, these frameworks are foundational to curriculum design with the goal of helping leaders approach problems holistically and iteratively, fostering a mindset geared towards continuous improvement and adaptation. Leadership development is viewed as a continuous journey rather than a one-time event. The curriculum supports long-term growth, allowing for leaders to evolve their skills as they progress through different stages of their careers and as organizational needs change.

Experiential Learning. Here, the focus lies on real-world projects and challenges that allow for leaders to apply systems thinking and gather evidence on their interventions and impact. Experiential learning places leaders in real or simulated situations where they must apply their knowledge and skills to solve real-world problems in real time. This method is effective because it extends beyond theoretical knowledge, compelling leaders to think on their feet, adapt, and apply systems and design thinking principles in practice. It also enables them to see the immediate impact of their decisions and actions, providing for powerful learning experiences through timely feedback. This type of evidence-based approach to learning and development ensures that leaders are not just applying concepts in theory but are also able to critically evaluate and learn from their actions in real-world contexts and in real time.

Continuous Feedback and Adjustment. Using data and feedback can aid in continuously refining leadership development strategies. The collected feedback should not only be used for personal development, but also to inform and motivate collective leadership development. This can mean updating the curriculum, introducing new learning modules based on emerging needs, or adjusting methodologies to better suit the learning styles and preferences of developing leaders. Ultimately, implementing systems design in leadership development requires fostering a culture that values learning, feedback, and adaptation. Organizations must encourage leaders to embrace experimentation, learn from failures, and view leadership development as an ongoing process and not a discrete event. Development is embedded in everyday life.

Implementation Consideration. To effectively implement these principles, organizations should consider customization and flexibility to tailor development to the specific needs of the leaders and their organizational context. This involves being flexible and adaptive in the design and delivery of leadership development initiatives. Furthermore, organizations should consider and encourage cross-functional collaborations. Encouraging collaboration across different parts of the organization helps to ensure that leadership development is integrated and aligned with overall organizational strategies and goals. In doing so, leveraging technology to facilitate learning, gather data, and provide feedback is paramount. This includes online learning platforms, data analytic tools, and digital collaboration resources.

Reconsidering leadership development based on systems design principles from an evidence-based perspective offers a more comprehensive and effective approach to preparing leaders for the complexities of modern organizational life. It emphasizes the importance of understanding and influencing the systems within which leaders operate, using data to inform decisions and fostering adaptability, resilience, and collaboration.

## 4. Practical Perspective on Leadership Development Systems

This section takes a more practical perspective on systems thinking in leadership development. Shifting the focus to how reconsidering leadership development based on systems design principles looks on a day-to-day level, we enter the realm of embedding leadership development into the fabric of organizational life [29,30]. This approach moves beyond formal programs or training, emphasizing continuous learning, feedback, and leadership practice within the daily workflow [31,32]. How can development as a leader manifest in everyday activities?

### 4.1. Integrating Learning into Daily Work

Microlearning, job rotation and stretch assignments, as well as reflection and journaling are various ways amongst a multitude of possible integrated learning initiatives for daily application [33,34]. Short, focused learning activities embedded into the workday allow for leaders to develop skills in real time. Enriched by providing opportunities for leaders to take on different roles or challenging projects that stretch their abilities and foster new competencies enhance experiential learning [35]. Encouraging leaders to reflect on their experiences and learnings daily or more regularly fosters self-awareness and insight.

### 4.2. Continuous Feedback Loops

Continuous feedback loops within the web of organizational life are best cultivated via frequent real-time feedback. Implementing tools and practices for immediate feedback from peers, subordinates, and supervisors on leadership behaviors and decisions is crucial for learning and development. One way to achieve this is to establish a habit of regular check-ins wherein the premise is to move beyond the traditional annual review to more frequent, informal feedback focused on development and not just evaluation [36].

### 4.3. Leveraging Technology

Utilizing digital tools and platforms for on-demand learning resources, peer collaboration, and tracking development goals is a way of horizontally and vertically embedding leadership development into ongoing organizational activities [37]. In addition, promoting the use of internal social networks for sharing knowledge, asking questions, and learning from the experiences of others promises avenues for democratizing leader development in organizations [38].

### 4.4. Cultivating a Supportive Culture

Cultivating a supportive culture through formal and informal mentoring and coaching relationships that provide guidance, advice, and support on a day-to-day basis helps to instill a leadership practice in the daily workflow. Small groups or communities of practice where leaders can share challenges, solutions, and learn from each other function as peer learning groups, fostering a learning organization [30].

### 4.5. Rewarding Growth and Learning

A successful leadership practice within organizations is based on recognition of effort and progress. As such, acknowledging and rewarding not just outcomes but also the effort and commitment of leaders towards personal growth and helping others develop is essential. Moreover, integrating development into performance metrics allows for leadership development to be an integral part of what is regularly measured, discussed, and rewarded in organizations. Focusing on how a leader has helped others develop as leaders themselves reinforces the importance of development as a foundation for enhancing individual and collective performance [39].

### 4.6. Tailoring to Individual Needs

Personalized development plans align with individual strengths, weaknesses, and career aspirations, offering adaptive learning pathways based on feedback and changing needs. This personalization ensures that leadership development is relevant and effective, supporting leader growth in alignment with an organization’s strategic goals.

### 4.7. Emphasizing Intangible Skills in Daily Interactions

Demonstrating and encouraging empathy, self-awareness, and effective communication in everyday interactions models so-called soft skills, which are indispensable to effective leadership. In addition, encouraging practices that build resilience, such as mindfulness, stress management, and viewing challenges as opportunities for growth assist in cultivating a resilient mindset [40].

In a day-to-day context, leadership development becomes less about structured programs and more about creating an environment where learning, feedback, and development are part of the natural workflow. This requires a shift in mindset from both leaders and the organization, where development is seen as a continuous process integrated with work, not separate from it. Organizations can foster this environment by providing the structures, tools, and culture that support ongoing learning and growth for both formal and informal leadership.

## 5. Discussion and Implications

An overarching question addressed in this article is how to make leadership development more effective. The data over years and even decades suggest that the consumers of leadership development initiatives, which are typically delivered as programs or similar one-off events, believe that there are significant shortcomings in the development of leaders and leadership. The importance of leadership development remains a critical concern for most senior leaders, yet there remains a large gap between the skills needed of future leaders and the present levels and types of skills they have acquired.

Instead of proposing how to completely rethink leadership development or offer suggestions on how to fix the programs that are typically used in leadership development, we argue for reconsidering things from a systems perspective. Simply put, we may have reached the limits of what leadership development programs can accomplish. There is a point of diminishing returns that has been reached.

One reason why programs no longer meet the needs (if they ever did) of many if not most organizations that invest in leadership development is that the world continues to become more complex at an increasingly rapid rate of change, whereas our tools used in development have not kept up with this complexity. The result is what some have termed the complexity gap between the adaptive challenges leaders are facing and their current levels of individual complexity in their sensemaking and reasoning capabilities [41]. A notable research question going forward is this: How might the adoption of systems thinking and the implementation of open systems design principles help reduce this noted complexity gap?

In answering this question, it is helpful to consider the nature of human development. How individuals develop is not an episodic or programmatic process, but one that is continuous and ongoing potentially over the entire lifespan [42]. Systems thinking recognizes the importance of continuous cycles of events that transform inputs in the form of energy and motivation into outputs to the environment leading to feedback that serves as new input into the ongoing developmental cycle. Open systems involve cycles of events that operate continuously [43]. A related research question that this point raises concerns the optimal number of cycles needed to bring about meaningful developmental changes in a leader or a broader collective. Furthermore, are these cycles the same in terms of their respective numbers across the level of analysis?

In contrast, leadership development programs are more like closed systems given that the interaction with the organizational environment is minimal with little in the way of ongoing feedback cycles. As result, any development is probably limited and not long lasting. There is the possibility of using a program as part of a leadership development system, but it should be treated as a single practice embedded in a set of interdependent developmental pathways (i.e., practices and processes) and not a stand-alone event.

The reconsidered role of providers of leadership development should be on helping individuals learn the key principles of system design and to use them in developing their own (bespoke) leader development systems. This may sound theoretical or abstract, but we suggest several tangible ways to start embedding systems thinking into ongoing, everyday leadership development. A key to leveraging the systems nature of these suggestions is to deliberately attend to their interdependencies and not think of them as separate practices.

Reconsidering leadership development in terms of systems rather than discrete programs might also be seen as an ongoing organizational process. It could be part of a broader change management initiative focused on organizational culture. An organization that has historically used various programs to develop leaders and leadership is unlikely to successfully move to an open-systems approach very quickly because the latter involves new ways of thinking about development and puts more of the impetus for development on the individual rather than having an organizational unit like HR assume that responsibility. A related, more holistic research question to consider is how to better align leadership development systems with change management processes [44,45].

In the literature on deliberately developmental organizations, the recommended approach is to embed development for everyone into the ongoing processes associated with shared work [39]. This is what those authors call an everyone culture, which is antithetical to the elite (and elitist) high-potential leadership programs adopted in many corporate organizations. A place to start is to view current practices with more of a systems perspective, identifying the various practices and processes currently used for leadership development but also identifying their interdependencies. A follow-up question is to consider (or reconsider) how to scale the developmental system in aiming a version of an everyone culture. Doing so would foster a deliberate attempt at democratizing leadership development [38].

## 6. Conclusions

The point was made earlier that it is ridiculous to expect different results from the same actions over time. We fear that most initiatives devoted to developing leaders and leadership are mainly tweaking at the margins in terms of what is offered. Evidently, there is a hope that some new practice can be added to a program that will provide the leadership solution [sic] that many claim to offer. It is time to reconsider how leadership development is designed and delivered. By thinking in systems terms, especially using open systems principles, there is an inherent holistic focus with particular emphasis on the interrelationships and interdependencies among its parts.

There is also the issue of time. There are no “quick fixes” when it comes to human development, including leader development. The evidence is clear that any meaningful and long-lasting change requires a commitment to dedicated practice over a long time. Expecting substantive personal or collective change after just a few days or weeks devoted to attending a leadership program is a fool’s errand. Without a deep reconsideration of how leader development is part and parcel to human development, especially adult development [3,42], we fear that there will be a continued focus on relatively short-term programs with increasingly diminishing returns.

It was nearly 35 years ago that the American statesman and educator John Gardner opined that when it comes to leadership development: “We can do much better. Much, much better” [46] (p. xix). We urge a reconsideration of how we design, deliver, and evaluate leadership development initiatives. It is time to let go of how we did things in the past and strive to do much, much better.

## Figures and Tables

**Figure 1 behavsci-14-00548-f001:**
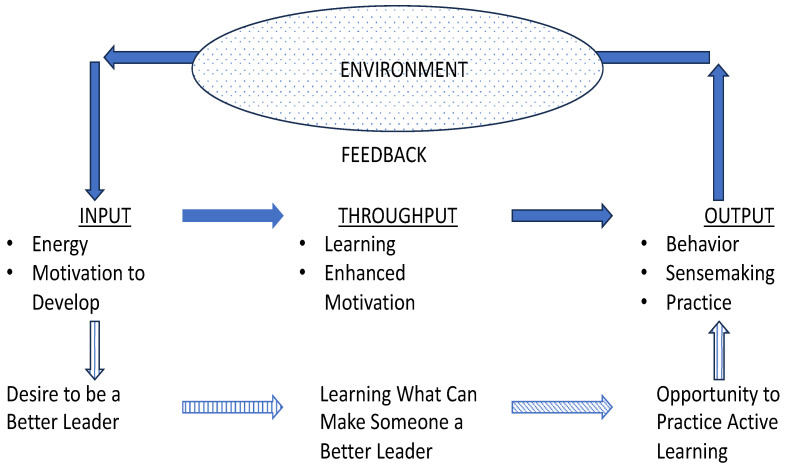
Leadership Development System.

**Table 1 behavsci-14-00548-t001:** Characteristics of Open Systems.

Characteristic	Brief Description
1. Importation of energy	Some form of energy is imported from the environment.
2. Throughput	Energy is transformed in some way within the system.
3. Output	Some type of product is exported to the environment.
4. Systems as cycles of events	There is a cyclical nature to the pattern of activities characteristic of energy exchange.
5. Negative entropy	Reverse or arrest the natural tendency toward death and destruction.
6. Information input, negative feedback, and the coding process	Information serves as a system input along with energy.
7. Steady state and dynamic homeostasis	There is a tendency to maintain some consistency and constancy in energy exchange.
8. Differentiation	Systems move in the direction of differentiation and elaboration.
9. Integration and Coordination	Processes counter differentiation to bring the system together for holistic functioning.
10. Equifinality	The same final state can be reached from differing initial conditions and through various pathways.

Note: Adapted from Katz and Kahn [7].

## Data Availability

No new data were created or analyzed in this study. Data sharing is not applicable to this article.

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
