# Peer review of "Reconsidering Leadership Development: From Programs to Developmental Systems"

_behavsci, 2024, doi:10.3390/bs14070548_

Round 1
Reviewer 1 Report
Comments and Suggestions for Authors
The abstract could be refined to capture the summary of the work better.
The introduction could provide the objectives, research questions and rationale to the study clearly and discuss them.
The methodology of the paper is not clearly presented and discussed, which is important for academic works.
The paper seems to be a literature review, it would be good to provide the methodological steps followed for the review at the end of the introduction and before the review is presented.
The review largely seems descriptive, it is recommended to showcase critical analysis, discussion and interpretations of the literature consulted.
The conclusion must highlight the general trends, patterns and gaps in the literature consulted.
Author Response
Thank you for your constructive comments. We have tried to clarify and elaborate on the focus and intent of our article, which is not an empirical study or a literature review. Instead, we present a scholarly essay arguing for the reconsideration of leadership development using principles grounded in open systems theory.
- We have revised the abstract to note the intended focus on open systems theory and related design principles.
- The article is more in the spirit of a scholarly essay arguing for a reconsideration of leadership development approaches based on open systems theory rather than program interventions, which we argue reflect (unintentionally) closed systems thinking. As such, this is not an empirical study; thus, there are no research questions or methodology to report. We have tried to be clearer about the objectives of our article in the introduction.
- This article is a scholarly essay. There is no methodology to report.
- We would not characterize this article as a literature review. We do not attempt to review the entire literature on open systems theory or design thinking. That would be a very different paper. Instead, our objective is to argue that the leadership development field has overlooked the importance of using open systems theory as a theoretical foundation for how to best develop leader and leadership.
- Again, we would not characterize this article as a literature review or a study. We describe relevant features of open systems theory and illustrate how they can be applied to potentially improve leadership development initiatives. Apologies, but we do not know how to interpret or respond to this comment: "...it is recommended to showcase critical analysis, discussion and interpretations of the literature consulted."
- Your comment that the discussion should highlight the general trends, patterns and gaps in the literature consulted would be appropriate for a more traditional literature review; however, that is not the intent of this article. That said, we have pointed out that most of the leadership development field is focused on programs as the best way to develop leaders and leadership. There is a gap in terms of including the ongoing interaction with the environment that is needed to develop individuals and collectives over the long term.
Reviewer 2 Report
Comments and Suggestions for Authors
Review – Behavioural Sciences
Reconsidering Leadership Development: From Programs to Developmental Systems
The justification for reconsidering leadership development is well explained in the introduction with supporting literature.
A different perspective drawing on Open Systems Theory is proposed and discussed beginning with a clear introduction of what this theory is about and how it can be applied to leadership development.
The discussion about what each characteristic (1-10) of Open Systems is presented in an accessible way and easily understood. It is good that this is presented before Table 1 which provides a brief summary. Figure 1: Leadership Development System shows how the theory is applied and highlights the cyclic process. I think people working in the leadership field would value this presentation of the characteristics of the theory where it shows what this would like in practice. The detail provided clearly indicates how this type of approach can meet the various leaders’ needs in different contexts with a focus on having continuous and ongoing inputs from the environments. This seems important when working in large organisations influenced by government priorities for example.
The key points/discussion highlighting systems principles will support understanding to progress leadership development drawing on a systems approach.
The authors take the principles from the theory and provide examples of how these could be applied in practice. This section 3 discussion will be particularly useful for readers working in the field of leadership development with considerations of the daily workflow.
I have no suggestions for change.
Author Response
Thank you for your supportive comments. We very much appreciate that you understand our approach is grounded in Open Systems Theory and that the focus in on leadership development not broader organizational systems.
Reviewer 3 Report
Comments and Suggestions for Authors
The article is properly structured, presenting a clear articulation in its different structures. It would be interesting if the conclusions presented a greater number of theoretical and practical examples of possible windows for future research.
Author Response
Thank you for your constructive and supportive comments.
We have revised the final section of the paper to include a few additional examples of research questions and practical applications to consider.
Reviewer 4 Report
Comments and Suggestions for Authors
In your introduction, you need to convince the reader that a) leadership development is currently not based on systems design principals and b) in what context are you situating leadership development: K-12 education? Higher education? Business? Medicine? Public sector leadership?
What is more, the first two paragraphs make claims without sources except for line 29, which is self-sourced. What kind of leadership development are you interested in? University-based? Licensure-based? Professional development for current leaders?
Page 3 begins a retelling of ten characteristics from an article written 46 years ago. It is unclear what new contributions this manuscript is trying to make.
The references are very old, with many of them not from leadership and not grounded in research. Part of the problem with this is that there is ample international, peer-reviewed research demonstrating the use of system design principles in leadership development. There is zero attention paid to more recent work on leadership development, for example:
Hopkins, M., & Woulfin, S. L. (2015). School system (re) design: Developing educational infrastructures to support school leadership and teaching practice. Journal of educational change, 16, 371-377.
Martens, B. R. (2011, September). The impact of leadership in applying systems thinking to organizations. In Proceedings of the 55th Annual Meeting of the ISSS-2011, Hull, UK.
Naicker, S. R., & Mestry, R. (2016). Leadership development: A lever for system-wide educational change. South African Journal of Education, 36(4), 1-12.
Nagahi, M., Hossain, N. U. I., Jaradat, R., Dayarathna, V., Keating, C., Goerger, S., & Hamilton, M. (2022). Classification of individual managers' systems thinking skills based on different organizational ownership structures. Systems research and behavioral science, 39(2), 258-273.
Generally speaking, I am recommending this manuscript be rejected because fundamentally it does not demonstrate that there is actually a lack of system design principles in use in leadership development, and the current work has not been given a context. In addition, the lack of new research and the use of self-citations from books and book chapters is problematic.
Comments on the Quality of English LanguageThe English is fine.
Author Response
- In your introduction, you need to convince the reader that a) leadership development is currently not based on systems design principals and b) in what context are you situating leadership development: K-12 education? Higher education? Business? Medicine? Public sector leadership?
Authors Response
Thank you for your suggestions. We make the point that programs are the typical approach used in developing leaders and leadership. There is lots of evidence for that – just check out the websites of prominent leadership development consultancies. There is no evidence that systems principles are the foundation for many leadership development interventions regardless of context. As for context, we are not limiting our comments to any one context domain. We believe that the points we raise apply across all contexts in which leadership development initiatives are undertaken.
- What is more, the first two paragraphs make claims without sources except for line 29, which is self-sourced. What kind of leadership development are you interested in? University-based? Licensure-based? Professional development for current leaders?
Authors Response
The reviewer appears to hold the position that all self-citations are bad or lack academic integrity. We disagree. This is from the Turnitin website (https://www.turnitin.com/blog/what-is-self-citation-and-what-does-it-have-to-do-with-academic-integrity): “Self-citation is an act of academic integrity when expanding on previous research or referring to previously published work. There are legitimate reasons to self-cite; work is largely, especially in scientific research, cumulative and it’s important to attribute prior foundational work.”
The self-cited articles and book chapters (except for the forthcoming book that has not yet been published) have received a total of 6,470 citations, or over 1,000 per work. We think that merits inclusion in the present article.
As for the “type” of leadership development that interests us, we believe our comments apply to leadership development in any context. There is no evidence to suggest that the principles of open systems theory are irrelevant in any of the contexts you mention.
- Page 3 begins a retelling of ten characteristics from an article written 46 years ago. It is unclear what new contributions this manuscript is trying to make.
It is a classic (i.e., highly cited) book, not an article. And its relevance is in terms of the theoretical grounding of leadership development initiatives. We elaborate on these 10 characteristics of open systems theory and, more importantly, their relevance to leadership development initiatives on pages 3-4 and summarize the principles in Table 1. We argue that these principles are not only relevant to improving initiatives designed to develop leaders and leadership, but they have been widely overlooked in the field.
- The references are very old, with many of them not from leadership and not grounded in research. Part of the problem with this is that there is ample international, peer-reviewed research demonstrating the use of system design principles in leadership development. There is zero attention paid to more recent work on leadership development, for example:
Hopkins, M., & Woulfin, S. L. (2015). School system (re) design: Developing educational infrastructures to support school leadership and teaching practice. Journal of educational change, 16, 371-377.
Martens, B. R. (2011, September). The impact of leadership in applying systems thinking to organizations. In Proceedings of the 55th Annual Meeting of the ISSS-2011, Hull, UK.
Naicker, S. R., & Mestry, R. (2016). Leadership development: A lever for system-wide educational change. South African Journal of Education, 36(4), 1-12.
Nagahi, M., Hossain, N. U. I., Jaradat, R., Dayarathna, V., Keating, C., Goerger, S., & Hamilton, M. (2022). Classification of individual managers' systems thinking skills based on different organizational ownership structures. Systems research and behavioral science, 39(2), 258-273.
Authors Response:
By very old, do you mean “classic?” We believe the classic texts on open systems theory have been overlooked in the leadership development field. Something else to consider is that recent does not necessarily mean better. We have tried to include a balance of more classic references with more recent ones. The evidence for scholarly impact is typically assessed in the form of citation counts and those take time to accrue. We think it would be a major mistake to focus on the recent to the exclusion of older works that have shaped the field, which we can estimate through article/book citation counts.
Related to impact, the “very old” book by Katz and Kahn (1978) has been cited more that 33,000 times. The references you provide have been cited collectively a total of 143 times. Impact matters. Also, the South African Journal of Education where the Naicker and Mestry (2016) article was published has a 2022-23 Journal Impact Factor of 0.87. That means that an article published in that journal received on average fewer than 1 citation in 2022. The Naicker and Metry article has received 61 citations, which makes it a bit of an outlier, although their focus was on a qualitative analysis of a single leadership program in a single school district. The Nagahi et al. (2011) piece published in a conference proceeding more than a decade ago has received a total of 6 citations. That is not very impactful. The other articles that you cite (a) are not focused on open systems theory, and/or (b) are not focused on leadership development.
We stand by our approach to include classic (and overlooked) literature rather than focusing on more recent and obscure references. Counter to your claim, there is much more than “zero attention paid to more recent work on leadership development” and a more objective review of our reference list would support that assertion. For example, the median publication date of the self-cited works in the article is 2015, and ranges from 2000 to 2025.
- Generally speaking, I am recommending this manuscript be rejected because fundamentally it does not demonstrate that there is actually a lack of system design principles in use in leadership development, and the current work has not been given a context. In addition, the lack of new research and the use of self-citations from books and book chapters is problematic.
Authors Response:
We appreciate your concerns but take exception to the claims of excessive self-citation. This is especially the case given you provide no alternative literature that has made similar points. You also do not present any evidence that the self-citations – including those from works in peer-reviewed journals -- are irrelevant to the arguments being presented here. We do not understand why this is “problematic” and you offer no explanation for this allegation. We take strong exception to your allegation because it goes directly to issues of academic integrity, something we take very seriously.
Round 2
Reviewer 4 Report
Comments and Suggestions for Authors
Despite the author's adamant response that self-citation and old/classic references are ok, it still doesn't address the fundamental problem with this work: there is no evidence it is providing something new to the field. Even if I were to totally concede that the reference lists are adequate (which I do not), the research itself does not make a novel contribution.
The danger of self-citation is that it does not allow the reader to see how the new work adds to the existing literature, including your own. You are correct that many of your references are considered "classics"-- however other research in this field has occurred in the years since. Because the reader is able to identify more recent research in this area, it undermines your introductory argument that little to no work exists in this field.
Author Response
The reviewer seems to have some sort of personal agenda regarding the authors and our work. The review is (again) unhelpful and ill-informed. The comments about self-citation make no sense given that the first author has been at the forefront of developing a scientific, evidence-based foundation to the field of leadership development for more than two decades. As for the potential contribution, the Editor and three other independent reviewers believe that there is a meaningful contribution worthy of publication in this journal. This reviewer is the only one who believes the manuscript does not make a publishable contribution yet fails to articulate clearly why that is the case.